# Empirical evidence of a fluctuation theorem for the wind mechanical power input into the ocean

Achim Wirth[1] and Bertrand Chapron[2]

[1]Univ. Grenoble Alpes, CNRS, Grenoble INP, LEGI, 38000 Grenoble, France
[2]LOPS, Ifremer, Plouzané, France

**Correspondence:** achim.wirth@legi.cnrs.fr

**Abstract.** The ocean dynamics is predominantly driven by the shear-stress between the atmospheric winds and ocean currents. The mechanical power input to the ocean is fluctuating in space and time and the atmospheric wind sometimes decelerates the ocean currents. Building on 24-years of global satellite observations, the input of mechanical power to the ocean is analysed. A Fluctuation Theorem (FT) holds when the logarithm of the ratio between the occurrence of positive and negative events, of a certain magnitude of the power input, is a linear function of this magnitude and the averaging period. The flux of mechanical power to the ocean shows evidence of a FT, for regions within the recirculation area of the subtropical gyre, but not over extensions of western boundary currents. A FT puts a strong constraint on the temporal distribution of fluctuations of power input, connects variables obtained with different length of temporal averaging, guides the temporal down- and up-scaling and constrains the episodes of improbable events.

## 1 Introduction

The exchange of heat, momentum and matter between the atmosphere and the ocean has a strong influence on our climate (Stocker et al. (2013)). Recent advances in satellite and in-situ based global Earth Observation (EO) systems and platforms, have significantly improved our ability to monitor ocean-atmosphere interactions. In the present work the exchange of momentum is considered. More precisely, we investigate the flux of mechanical power into the ocean mixed-layer at the ocean surface. It is caused by the shear-stress at the surface due to the difference between the atmospheric winds and the ocean currents near the surface, in the corresponding planetary boundary layers. Various physical processes occurring at the air-sea interface on a large range of scales in space and time are important for the momentum transfer. For a general discussion on air-sea interaction we refer to Csanady (2001) and Veron (2015)), for ocean energetics to Ferrari and Wunsch (2009) and for wind work to Wunsch (1998). The atmospheric winds are usually stronger than the ocean currents and therefore the atmosphere mostly loses energy at the interface by friction and the ocean mostly gains energy. As a feedback mechanism, the presence of surface currents will then modulate the air-sea transfer of momentum (Bye (1985), Renault et al. (2017)). The energy exchange is not conservative

and most of the mechanical energy is dissipated (Duhaut and Straub (2006), Wirth (2018), Wirth (2019)). In the present work we are not concerned with the details of the exchange in the respective boundary layers (see e.g. Veron (2015)) but suppose that it is well represented through bulk formulas of air-sea interaction (Fairall et al. (1996)). In those models the power input is estimated based on the shear-stress at the surface and the ocean current near the surface and also depends on the sea state and

the density stratification in the atmosphere and the ocean.

More precisely, we consider the mechanical energy exchange between the atmosphere and the ocean at a time, $t$, over a fixed surface area $A$ of the ocean (the area which spans $10^o$ in the longitudinal and the latitudinal direction) . For this area we evaluate the mechanical power the ocean gains at the interface $\mathcal{P}(t)$. Due to the turbulent dynamics in the atmosphere and the ocean the quantities are fluctuating over a large range of scales in time and space.

We focus on two properties of the mechanical power input to the ocean at the surface: (i) on average the ocean gains energy at the interface $\langle \mathcal{P}(t) \rangle > 0$ (where $\langle . \rangle$ represents an average over the observation period and several surface areas $A_i$) and (ii) the power input is fluctuating, in time and space, due to the turbulent motion in the atmosphere and the ocean and negative events, with $\mathcal{P}(t) < 0$, occur.

Today, fluctuations are the focus of research in statistical mechanics, which was traditionally concerned with averages.

Fluctuations in a thermodynamic system usually appear at spatial scales which are small enough so that thermal, molecular, motion leaves an imprint on the dynamics as first noted by Einstein (1906) (see also Einstein (1956) and Perrin (2014)). The importance of fluctuations is, however, not restricted to small systems. Fluctuations can leave their imprint on the dynamics at all scales when (not necessarily thermal) fluctuations are strong enough.

Turbulent fluid motions are typical examples (i.e Frisch (1995)), for which average motions can not be understood or mod-

elled without some knowledge about the turbulent fluctuations, and vice versa . Turbulent fluctuations can be especially pronounced in geophysical flows, which are highly anisotropic due to the influence of gravity and rotation. This leads to a quasi two-dimensional dynamics and an energy cascade from small to large scales and strong fluctuations (see i.e. Boffetta and Ecke (2012) for a review on 2D turbulence). Likewise, the description of air-sea interactions on large time scales may not be understood without some knowledge of the fluctuations at smaller and faster scales. Furthermore, the research interest in many

natural systems lies also in the fluctuations not only in an average state, weather and climate dynamics are examples where we focus on the fluctuations of the same system on different time scales. For the weather the time scale of interest is from roughly an hour to a week, for the climate the focus is from tenths to thousands of years and beyond.

A recent concept which is presently subject of growing attention in non-equilibrium statistical mechanics are Fluctuation Theorems (FT) (see e.g. Evans et al. (1993), Gallavotti and Cohen (1995a), Gallavotti and Cohen (1995b), Ciliberto et al.

(2004), Shang et al. (2005) and Seifert (2012)). Not only the average values of quantities like entropy, work, heat or other, are studied, but their fluctuating properties are scrutinised. There are different forms of FTs, reviewed in detail by Seifert (2012). In the present paper we focus on the FT put forward in Gallavotti and Cohen (1995a), Gallavotti and Cohen (1995b) and Gallavotti and Lucarini (2014), corresponding to the detailed fluctuation theorem in the limit of large averaging times. When the FT applies to a fluctuating quantity, as i.e. $\mathcal{P}(t)$ in the present study, it relates the probability to have a negative event, i.e.

the ocean loses energy, to the probability of a positive event, i.e. the ocean gains energy, of the same magnitude. The FT is not

concerned with instantaneous values but considers the fluctuations of temporal averages over varying averaging time. The FT, which is stated precisely in the next section, thus puts a strong constraint on the fluctuations of the quantity considered and its temporal averages of varying length.

FTs have been established analytically for Langevin type problems with thermal fluctuations (Seifert (2012)). Most experimental data comes also from micro systems subject to thermal fluctuations. The thermodynamic frame of the quantities considered, as entropy, heat and work is not necessary to establish FTs. Examples of non-thermal fluctuations are the experimental data of the drag-force exerted by a turbulent flow (Ciliberto et al. (2004)) and the local entropy production in Rayleigh-Bénard convection (Shang et al. (2005)). For these non-Gaussian quantities the existence of a FT was suggested empirically. Our work is strongly inspired by these investigations of the FT in data from laboratory experiments of turbulent flows.

In Wirth (2019) the FT was investigated for three parameterizations of air-sea interaction and we refer the reader to this work for the theory and analytical solutions on fluctuating air-sea interaction in these idealised models. In that publication the concept of FT is also placed in a broader context of fluctuating dynamics and the relation to the fluctuation-dissipation-relation and the fluctuation-dissipation-theorem is given (see also Seifert (2012) for a general discussion). Here we extrapolate the research of Wirth (2019) by applying the concept of FTs to data derived from satellite measurements and discuss their relevance. It is important to notice that even in the case of the idealised models the FT was not established by analytical calculation, but it was confirmed numerically that the FT is obtained asymptotically, in the long-time limit, when the averaging time is larger than the time-scale of the slow ocean-dynamics.

## 2   The Fluctuation Theorem

We are interested in the mechanical power, $\mathcal{P}(t)$, absorbed by the ocean over a given surface area, $A$, of the ocean surface and an observation period $t_{\mathrm{obs}}$. We suppose that $\mathcal{P}(t)$ is a statistically stationary random variable, meaning that its statistical properties (mean value, moments and temporal correlations) do not change when shifted in time. Its statistical properties, at every instance of time, are completely described by its probability density function (pdf), $p(z)$, which gives the probability that $\mathcal{P}(t)$ takes values between $z_1$ and $z_2$ by integration: $\Pr[z_1 < \mathcal{P}(t) < z_2] = \int_{z_1}^{z_2} p(z)dz$. The symmetry function is:

$$S(z) = \ln\left(\frac{p(z)}{p(-z)}\right). \tag{1}$$

It compares the occurrence of events when the ocean receives power of magnitude $z$ to the occurrence when the ocean loses power of the same magnitude. We further denote the normalised energy received during an interval $\tau$ starting at time $t_0$, by:

$$\overline{E(t_0)}^{\tau} = \frac{\int_{t_0}^{t_0+\tau} \mathcal{P}(\tau')d\tau'}{\int_0^{t_{\mathrm{obs}}-\tau} \int_{t_0}^{t_0+\tau} \mathcal{P}(\tau')d\tau'dt_0/(t_{\mathrm{obs}} - \tau)}, \tag{2}$$

where $t_{\mathrm{obs}}$ is the total length of the available data record. The corresponding pdf is denoted by $p(z, \tau)$ and the symmetry function by $S(z, \tau)$. Note that the averaging starts at time $t_0$ and extends over the interval $\tau$.

The Galavotti-Cohen fluctuation theorem (called FT in the sequel for brevity) holds for $\mathcal{P}$ if, for averaging times larger than a convergence time-scale of the FT ($\tau \gg \tau_0$), two conditions are satisfied: (i) the symmetry function depends linearly on the

variable $z$, and (ii) on $\tau$:

$$S(z,\tau) = \sigma\tau z, \tag{3}$$

where $\sigma$ is called the contraction rate. The contraction rate $\sigma > 0$ (see Gallavotti and Cohen (1995a), Gallavotti and Cohen (1995b), Ciliberto et al. (2004) and Shang et al. (2005)) depends on the problem considered. We did not manage to determine it from observed quantities. In systems where the fluctuation is due to thermal motion its value is related to the thermal energy, that is the product of the Boltzmann constant and temperature, $k_B T$. When fluctuations arise from turbulent motion the temperature has (almost) no influence on the fluctuations and the contraction rate $\sigma$ depends on the turbulence. Indeed, in the incompressible Navier-Stokes equations temperature does not appear explicitly and only the kinematic viscosity has a slight dependence on temperature. There is, therefore no reason why $k_B T$ is a governing parameter of the problem.

If the FT holds it is sufficient to know the probability for either $z > 0$ or $z < 0$ to obtain the whole pdf, when $\sigma$ is known. The FT therefore constraints "half" of the pdf, a strong constraint in the absence of an equivalent of the Boltzmann distribution. This property also allows to calculate the probability of the rare events of $z < 0$ from frequent events $z > 0$.

For a dynamical system the FT may or may not hold and it might only be valid for a range of values. It was already noted in Gallavotti and Cohen (1995a) and Gallavotti and Cohen (1995b) that the FT might only be valid for values $z < z^*$, when the large deviation function (see i.e. Touchette (2009)) diverges outside the interval $[-z^*, z^*]$. More recently it was recognised that boundary conditions, that is the value $\mathcal{P}(t)$ at $t = t_0$ and $t = t_0 + \tau$, can leave their signature in the symmetry function $S(z,\tau)$, even when the limit of $\tau \to \infty$ is taken, whenever the pdf $p(z)$ has tails which are exponential or less steep than exponential (see Farago (2002), Van Zon and Cohen (2004) and Rákos and Harris (2008)). In such case an extended FT (EFT) should be expected, which shows a linear scaling of the symmetry function near the origin with a transition to a flatter curve for larger values. An analytic expression of the symmetry function, or the value of $z^*$ is obtained only for very idealised cases and the results presented here are empirical.

## 3 Power Input

The calculations of the power input to the ocean are based on the shear-stress at the surface and the ocean velocity. The shear-stress is usually evaluated, based on the difference between the horizontal wind velocity $\mathbf{u_a^s}$, usually taken at 10m above the ocean surface and the horizontal ocean surface-current $\mathbf{u_o^s}$, using the quadratic drag law (see i.e. Renault et al. (2017)):

$$\mathbf{F} = C_d \sqrt{(\mathbf{u_a^s} - \mathbf{u_o^s})^2}(\mathbf{u_a^s} - \mathbf{u_o^s}). \tag{4}$$

The drag coefficient $C_d$ depends on the sea-state and the stratification in the atmosphere and the ocean, it is obtained using bulk formulas (Fairall et al. (1996)). Variations of the drag coefficient are not considered and all the results are independent of a constant $C_d$.

To obtain the power input, the vector product between the shear-stress and the ocean current-velocity is taken:

$$\mathcal{P}(t) = \mathbf{F} \cdot \mathbf{u_o}. \tag{5}$$

For the work done on the large-scale geostrophic-circulation, Wunsch (1998) and Zhai et al. (2012) used the surface geostrophic velocity estimates from altimetry for $\mathbf{u_o}$. Using model data, Rimac et al. (2016) chose the velocity at the surface to calculate the total power input, to then evaluate that only a fraction of this power is transmitted to the interior ocean at the base of the mixed layer. In the present work, largely building on 15-m drogued drifter velocities (Rio et al. (2014)), we use for $\mathbf{u_o}$ the estimation of the current velocity at 15m depth. By taking the velocity at 15m rather than at the surface we exclude the power that is promptly dissipated by viscosity in the Ekman-spiral, but include the power that is supplied to the ageostorpic and geostropic dynamics at the core of the mixed layer.

## 4 Data

In this study, we build on the newly released GlobCurrent products, now available via the Copernicus Marine Environment Monitoring Service (CMEMS, http://marine.copernicus.eu/services-portfolio). Essentially building on the quantitative estimation of ocean surface currents from satellite sensor synergy, the production has been performed of a 25 years reanalysis of global, $1/4^o$ maps of ocean currents at two levels, the surface and 15m depth. The data is obtained from the combination of altimetry, GOCE, wind and in-situ data (largely building on 15-m drogued drifter velocities) (Rio et al. (2014)).

Strongly based on altimeter data, this global ocean surface current product, and also similar global observation-based products (Bonjean and Lagerloef (2002), Sudre et al. (2013)), suffer from well-known limitations. The full spatio-temporal ocean dynamics is certainly not well captured, possibly missing part of the geostrophic component and a number of dominant ageostrophic signals (e.g. inertial oscillations). Also, accuracy is strongly reduced in the Equatorial Band where the geostrophic approximation fails, in coastal areas where altimetry accuracy decreases and where ageostrophic currents often dominate, and in the seasonally ice-covered Polar Seas. Nevertheless, this global ocean surface current product provides a consistent data set covering the last 25 years.

Satellite winds are from the Copernicus project (http://marine.copernicus.eu/services-portfolio/access-to-products/). They are from scatterometer and radiometer wind observations. It is a blended product based on the different missions (ERS-1, ERS-2, QuikSCAT, and ASCAT) available at $1/4^o$ spatial resolution and every 6 hours and is described in Bentamy et al. (2017) and Desbiolles et al. (2017). The data record for which wind and current data is available extends over 24 years, 1993–2016, at the same resolution in space and time.

The FT is a property that concerns the tails of a pdf, and it is necessary to consider a large amount of data, as provided by the GlobCurrent products. Still, a time record of 24 years of data coverage at a single location is too small for empirically suggesting or refuting the existence of a FT. To increase the amount of data, we use different tiles $A_i$ that obey similar statistical properties. The tiles represent an effective area of $0.5^o$ in the longitudinal and latitudinal direction. For a trade-off between ensemble size and similar statistical properties, we choose to consider domains extending $10^o$ in the longitudinal and latitudinal direction, composed of $20 \times 20$ non-overlapping tiles each.

Four domains are considered, the first is in the recirculation area of the subtropical gyre ($20^o - 30^oN$, $20^o - 30^oW$) of the North Atlantic (case: ASG), the second in the Gulf Stream extension ($35^o - 45^oN$, $35^o - 45^oW$)(case: GSE). The third is in

the recirculation area of the subtropical gyre of the North Pacific ($15^o - 25^o N$, $150^o - 160^o E$)(case: PSG) and the fourth in the Kuroshio extension ($30^o - 40^o N$, $150^o - 160^o E$) (case: KUE). The data record from which wind and current data is available extends over the 24 years, 1993–2016, at a resolution of 6h. At four occasions in time data was missing. The gaps were filled by linear interpolation.

The pdfs of $\overline{E(t)}^\tau$ are calculated for an interval that spans at least twice the standard deviation of each pdf from the origin. Note, that the average is unity by definition. The pdfs are calculated with three different resolutions (bin sizes). The interval is separated into 21, 31 and 41 bins of equal size and the pdfs are obtained by counting the number of occurrences for each bin. The symmetry function is only calculated when probabilities are lager than $10^{-3}$ per bin, this led to an omission of bins in case ASG, only.

## 5   Results

The pdfs $p(z, \tau)$ for the four domains and for different values of averaging times $\tau$ are presented in the left panels of figs. 1, 2, 3 and 4. All clearly display non-Gaussianity.

    With increasing averaging period, the pdfs become more centred around unity, which is the average value, see eq. (2). This can be verified by comparing the graphs for $\tau = 125, 250, 500$, and $1000$ days. It is a consequence of the central limit theorem
and occurrences of negative values become less likely for larger $\tau$. In the right panels of figs. 1, 2, 3 and 4 the symmetry function divided by the averaging period is plotted. These plots are similar to those in Gallavotti and Lucarini (2014), who verified the FT in an idealised numerical model.

    The verification of the FT, that is of eq. (3), is two-fold. First, we verify the linear dependence of the symmetry function on $z$ for different averaging periods $\tau$, and determine the slope. Second, we verify that the slope is a linear function of $\tau$ for times
20 larger than the convergence time-scale of the FT, $\tau > \tau_0$, of the system. That is, we first have to confirm that the lines in the right panels of figs. 1, 2, 3 and 4 converge towards straight lines for increasing averaging periods and second we see if the lines superpose when increasing averaging periods. This is demanding, and a large amount of data is necessary. For the first point, we have to consider the pdf for an extended range in $z$, including the tails, asking for ensemble sizes (number of intervals of length $\tau$) large enough so that we can observe a clear scaling behaviour. For the second point, we have to increase $\tau$ to verify
convergence. Furthermore, for larger $\tau$ the pdfs are more and more peaked around unity and negative events become less and less likely.

    For the four domains, we observed a convergence of the normalised symmetry function with increasing averaging time. This indicates the existence of a large deviation principle (see i.e. Touchette (2009)). For the domains within the recirculation area (ASG and PSG) of the subtropical gyre a convergence towards a linear variation with $z$ is observed in less than $\tau_0 \approx 1$ year.
This points towards the existence of a FT, as both points put forward at the beginning of the previous paragraph are observed. For the extensions of the western boundary currents (GSE and KUE) , the convergence does not achieve a linear behaviour of the normalised symmetry function. This shows that a FT does not hold, as the first point put forward at the beginning of the previous paragraph is not satisfied.

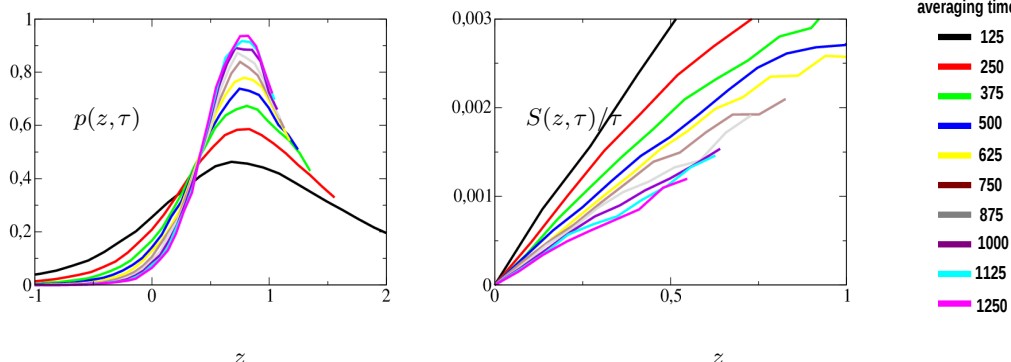

**Figure 1.** The pdf $p(z,\tau)$ (left) and the symmetry function normalised by the averaging time $S(z,\tau)/\tau$ (right) as a function of $z$ for different $\tau$ (see caption). The variable $\tau$ gives the length of the averaging interval in terms of observations done every 6 hours. Data are for case ASG, res $0.5^o$, 1993–2016, res 6h

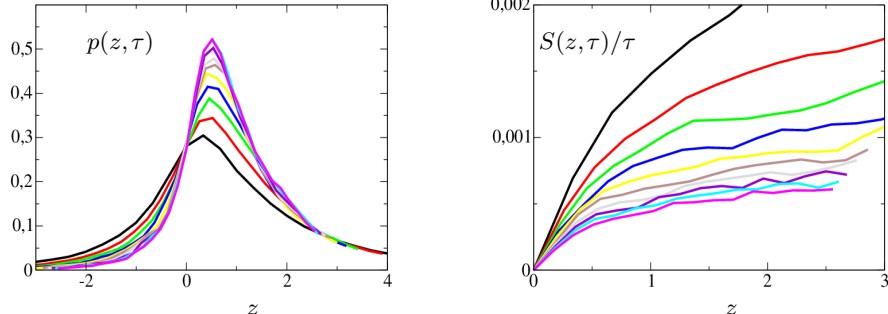

**Figure 2.** The pdf $p(z,\tau)$ (left) and the symmetry function normalised by the averaging time $S(z,\tau)/\tau$ (right) as a function of $z$ for different averaging times $\tau$ (see caption and legend of Fig. 1); data are from GSE, res $0.5^o$, 1993–2016, res 6h

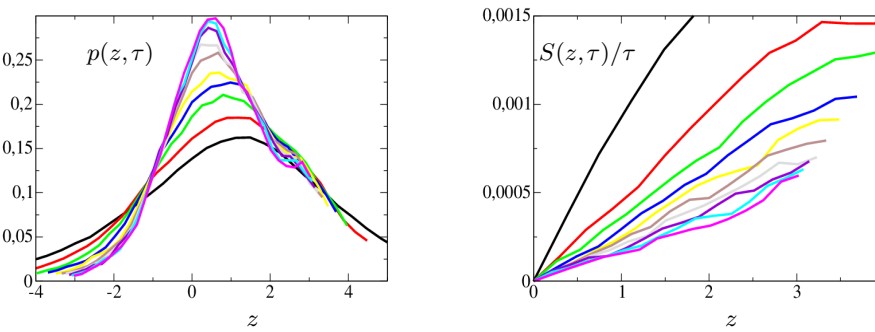

**Figure 3.** The pdf $p(z,\tau)$ (left) and the symmetry function normalised by the averaging time $S(z,\tau)/\tau$ (right) as a function of $z$ for different averaging times $\tau$ (see caption and legend of Fig. 1); data are for case PSG, res $0.5^o$, 1993–2016, res 6h

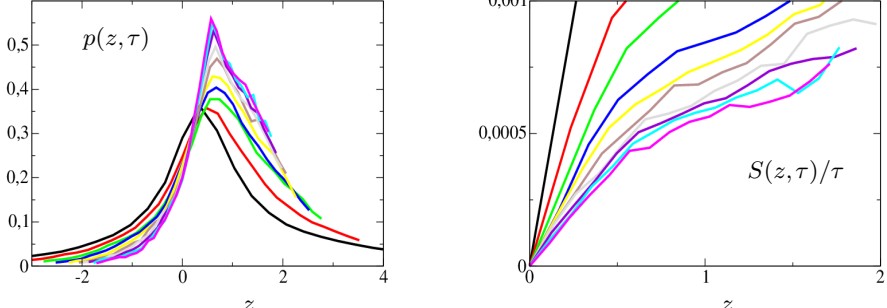

**Figure 4.** The pdf $p(z,\tau)$ (left) and the symmetry function normalised by the averaging time $S(z,\tau)/\tau$ (right) as a function of $z$ for different averaging times $\tau$ (see caption and legend of Fig. 1); data are for case KUE, res $0.5^o$, 1993–2016, res 6h

| exp. | ASG | GSE | PSG | KUE |
|---|---|---|---|---|
| $\gamma$ (21 bins) | 1.11 | 3.55 | 0.90 | 1.83 |
| $\gamma$ (31 bins) | 1.02 | 3.63 | 0.92 | 1.71 |
| $\gamma$ (41 bins) | 0.92 | 3.54 | 1.03 | 1.75 |

**Table 1.** Index $\gamma$ measuring the alignment of the normalised symmetry function for the four experiments and different resolutions of the pdfs (number of bins).

The contraction rate $\sigma$ is the slope of the curves in the right panels of figs. 1, 2, 3 and 4. To estimate the alignment of the points for $\tau = 1250$ days, we constructed an index $\gamma$: the slope of the normalised symmetry function from the origin to the first bin divided by the slope from the origin to the last bin. A value $\gamma = 1$ indicates a perfect alignment of the first bin with the last. The index is presented in table 1 for the four different domains and three different resolutions of the pdf. For the recirculation
area of the subtropical gyre cases, the index varies around unity for the different bin sizes. It is significantly greater than unity in the Gulf Stream and the Kuroshio extension for all bin sizes considered.

We did not attempt to present error-bars in the figures and numbers in the tables, as uncertainties depend on the number of statistically independent events, that is the correlation time. In the case of air-sea interaction there are correlations due to the atmospheric dynamics (mostly synoptic), the ocean dynamics, the annual cycle, interannual variability and a climatic trend.
How these processes contribute to the tails of the pdf's, to improbable events, is currently a hot topic in climate science (see *i.e.* Ragone et al. (2018)).

## 6   Discussion

We obtain evidence that a FT applies to data within the recirculation area of the subtropical gyre in the Atlantic and the Pacific Ocean. In these cases the FT can be used to estimate the occurrence of rare negative events from frequent positive events of
the same magnitude for all averaging periods $\tau$ (measured in days). If the FT applies, the probability of the rare improbable negative events can be calculated from frequent positive events. The rare events when the ocean looses energy have recently been the focus of dedicated research (see e.g., Zhai et al. (2012), Wirth (2021)). Improbable negative events also lead to strong transfer of energy to small-scale turbulence in the atmospheric and oceanic boundary layers, potentially causing strong mixing in the atmosphere and ocean. Improbable events are often key for the system in a variety of applications and are the focus of
recent research in climate science (Ragone et al. (2018), Seneviratne et al. (2012)). As an example: in the Atlantic subtropical gyre (ASG) case the slope of the symmetry function is $S(z,\tau) = 2 \cdot 10^{-2} \tau z$, this means that an event of the magnitude $z = -1$ is $p = \exp(-2 \cdot 10^{-2} \tau)$ less likely than an event having the average value ($z = 1$) and an event of the magnitude $z = -2$ is $p = \exp(-4 \cdot 10^{-2} \tau)$ less likely than an event having twice the average value ($z = 2$). The variable $\tau$ gives the length of the averaging interval in terms of observations done every 6 hours, that is $\tau = 400$ corresponds to a period of 100 days. A FT
represents a tool to obtain the rare negative events from frequent positive events for all averaging times $\tau > \tau_0 \approx 1$ year and demonstrates that, to leading order, the probability of negative events vanishes exponentially with the averaging time.

The FT does not seem to apply in the highly non-linear Gulf Stream extension for $z \gtrsim 0.3$ and Kuroshio extension $z \gtrsim 0.5$. For these regions, the symmetry function follows a FT for small values of $z$, before the curve flattens. This resembles the behaviour found in the EFT (see section 2). Indeed, in these two cases (GSE & KUE) the tails of the pdf of $\mathcal{P}$ show pronounced super exponential tails and boundary values might be important leading to a behaviour predicted by an EFT. Nevertheless, a similar change of slope was also found using highly idealised models of air-sea interactions (discussed in Wirth (2019)), in which a friction term was added to the ocean. This suggests that other processes than air-sea interaction dominate in the extension of the boundary currents leading to a departure from a FT symmetry. When the scaling of the symmetry function flattens for higher power-input, the manifestation of a negative improbable event, versus a positive event of the same magnitude, becomes more likely.

During data analysis, we also found that a FT does not apply when islands or coastlines are present (not shown here). Departure from a FT for the power input to the ocean is found where horizontal dynamics dominates over the vertical ocean-atmosphere momentum exchanges. The influence of the horizontal transport of energy with respect to the injection of energy through the surface decreases with domain size considered, as the circumference of a domain grows linearly, whereas its surface growth is quadratic. Yet, determining the existence of a FT for larger ocean domains asks for more data, which is currently not available. Our results are purely empirical, a theory explaining why the power input follows a FT in some cases and not in others, is still missing.

Finally, we put the theory of FTs in the more general context of climate dynamics. A measurement, especially when coming from satellites always contains some averaging in space and time. A FT, when it applies, will help to relate averages over varying periods and is a powerful tool to guide the up and down-scaling of observational data in time and obtain the statistical information on shorter and longer time scales, which are not explicitly observed. More precisely, when the pdf of the power supply, and therefore also the symmetry function is known form observations for given averaging times the symmetry function can be calculated for shorter and larger averaging times and constrains "half" of the pdf. This is useful in down-scaling and the construction of statistical parameterizations of not directly observed dynamics over shorter time scales. On the other hand, the information can be useful for developing models for the persistence of events over large time-scales not yet observed. A FT can help to decide if the persistence in time of a phenomena is within the likeliness of the statistically stationary dynamics or due to external influences. Furthermore, when data from observations follow (or not) a FT, model data should do likewise. As such, the FT becomes a tool of investigating the fidelity of models.

Recently there is an increasing interest in the variability of atmosphere and ocean dynamics and in the exchange of the two, also at high frequency and their contribution toward the lower-frequency air-sea momentum and energy fluxes (e.g., Zhai et al. (2012), Wirth (2021)). It was found that higher frequency wind forcing increases the mixed layer depth (Zhou et al. (2018)). There is evidence that long term variability of the atmosphere ocean system as the Madden-Julian Oscillation and El Niño-Southern Oscillation, are influenced by higher frequency wind forcing (e.g., Bernie et al. (2007), Terray et al. (2012)). FTs when they apply give a connection between events averaged over different lengths of time and can help to evaluate the impact of not explicitly resolved scales. When they do not apply they might help to identify a specific, non stochastic, mechanism responsible.

We conclude by looking at our results from the stand point of dynamical systems. Statistical mechanics of systems in equilibrium are described by the Boltzmann distribution, which is completely determined by the temperature. In non-equilibrium statistical mechanics no such universal distribution is known (see i.e. Derrida (2007), Touchette (2009) and Frisch (1995)), but some quantities in some processes seem to follow a FT which constraints the pdf and might indicate some universality. The
mechanical power-input to the ocean by air-sea interactions, as a forced and dissipative dynamical system, may thus belong to a class of particular non-equilibrium systems exhibiting a FT symmetry property and offer guidance for climate studies.

*Author contributions.* AW has performed the coding, writing was shared by both authors

*Competing interests.* No competing interest

*Data availability.* Data is available under: http://marine.copernicus.eu/services-portfolio/access-to-products/ and http://marine.copernicus.eu/services-
portfolio)

*Acknowledgements.* This work was funded by Labex OASUG@2020 (Investissement d'avenir - ANR10 LABX56). These data were provided by the Centre de Recherche et d Exploitation Satellitaire (CERSAT), at IFREMER, Plouzane (France) and CMEMS. Part of this work was performed when AW visited LOPS, Brest. We are grateful to Abderrahim Bentamy for explanations concerning the data and Mickael Accensi and Jean-Fancois Piolle for help with the data analysis.

*Data availability.* Data is available under: http://marine.copernicus.eu/services-portfolio/access-to-products/ and http://marine.copernicus.eu/services-
portfolio)

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
