# Peer review of "Empirical evidence of a fluctuation theorem for the wind mechanical power input into the ocean"

_Nonlinear Processes in Geophysics, 2020_

## Referee Comment (RC1) · Anonymous Referee #1 · 20 Oct 2020

General comments:

This study investigates empirically whether or not the time integrated input of mechanical power from the atmosphere to the ocean obeys a fluctuation theorem. If this were the case, observations of the very common case where momentum is transferred from the atmosphere to the ocean could be used to infer probabilities for the rare opposite case. The paper is overall well-written and easy to follow, even if the reader is not closely familiar with ocean dynamics or fluctuation theorems. The core idea is sufficiently interesting for publication in this journal and constitutes a natural next step after the first author's previous study of conceptual models (Wirth 2019). The results

appear to be somewhat inconclusive but this fact alone should not exclude the paper from publication. I am mainly concerned with the data analysis in section 5 which is not very clearly presented, both in terms of the methodology and the actual discussion and plots.

Specific comments:

p.4 l4-5 "fixed surface area" this is probably not very important but is the surface area actually fixed when the sea state can change over time? If you always consider fixed geographical regions, wouldn't calm conditions lead to a smaller surface area than rough seas?

p.2 l22 "the focus" please make it clear whose focus you mean (the focus of most current research?)

p.2 l33-34 "not only concerned with instantaneous values" if I understand correctly, eq. 3 doesn't refer to instantaneous values at all, right? In that case you should cut "only" here.

p.3 l30 please make it unambiguous that the limit of large tau relates to both conditions and not just (ii). Also this is the first instance where tau_0 occurs, please explain what this refers to.

p.5 l27f consider including a map of the world showing these four regions to give non-oceanographers at least some idea where they are located, how large they are and what factors might influence the different dynamics.

p.5 l27f do you have some idea how sensitive your results are to the specific choice of your domains?

p.6 l1 what exactly do you mean by "an interval that spans twice the mean value [...] from the origin"? 0 +/- 2*mean( E_tau ) ? In that case why is zero not at the center of the left parts of Fig.1-4 ?

Figures 1-4: Please add axis labels to both parts of the figures. Then the captions of Fig. 2-4 don't need to repeat that of Fig.1, "as Fig.1 but for case XY" would be sufficient. Please give the unit of the averaging time as well.

p.6 l11you state that you will verify Eq. 3 in two steps so the reader expects these two to be addressed in order. It is however unclear to me which of the following two paragraphs is supposed to refer to which aspect (see further comments below).

p.6 l12 you claim that you "determine the slope" but that that slope is never actually shown or discussed directly. Why not fit lines to your curves and show us the estimated slopes (see comment below)? In that way we could also compare whether or not the slope differs between the regions which is hardly possible by comparing curves in different plots with different y-axes.

p.6 l13 you again mention tau_0, can you at least give some rough estimate how long that time-scale might be, relative to the length of your time series? Could this be inferred from the power-spectrum of the time-series?

p.6 18f "This indicates the existence of a large deviation principle" isn't it more important that this convergence is predicted by the FT? What is the relationship between the existence of an LD principle and a FT? Also is this the first or the second part of the verification mentioned above?

p.6 l19f "extension of the domains within ...", "extension of the western boundary current" please refer to the different regions by the acronyms you established before and also refer to the figures in which these results are shown.

p.7 l1f I'm not sure why you chose to quantify the linearity of your curves by this specially designed index. If I understand correctly, the scaled symmetry functions corresponding to long averaging times should be linear across the whole range of z-values. Why not simply fit a line via least squares to calculate the overall slope? Use $R^2$ to get an idea of the goodness of fit and plot the slopes against tau to observe the convergence behavior. I understand that the statistical interpretation in terms of confidence intervals is questionable but I don't see why your index is more appropriate. Unless I misunderstood your definition, there are many non-linear curves for which gamma=1.

p.8 l7 "extreme events are often key" of course extreme events in general are interesting but your framework doesn't describe just any kind of weather extreme but specifically unusually small (negative) values of atmosphere-ocean momentum transfer. Can you explain a bit more specifically why a rare event wherein the wind in the atmosphere is sped up by the ocean is of interest?

p.8 l9f I like this example, perhaps it would be even more illustrative if you put in actual numbers for tau? Say one month or one year? This, however raises the question how large tau has to be for the FT to hold ...

p.8 l12 "all averaging times" if I understand correctly, your FT only makes statements about long averaging times, right?

p.9 l3 "exp2 &4" please refer either to the figures or the abbreviations of the different regions in a consistent manner, the terms "expN" were never explicitly introduced.

p.9 l18 "guide the up and down-scaling" can you either give a reference for this claim or explain a little more how the FT could help with that?

Technical corrections:

p.2 l14: case mismatch between "the importance [...] is, [...] their imprint", please re-formulate

p.2 l17-18: the sentence with "can not be understood or modelled" is repeated verbatim, please cut or re-formulate.

p.2 l32: replace "i.e." by "e.g."

p.4 l7: replace "is" by "should be"

p.5 l6f "the production has been performed of ..." confusing sentence, do you mean "a near real-time data set, as well as a 24 year reanalysis, [...], have been produced" ?

p.5 l15 25 or 24 years ?

p.5 l20 "6h in time and 1/4° in space" this is repeated from the previous sentence.

p.5 l24 ", For" either change to lowercase or start a new sentence

p.5 l30 "from" instead of "form"

p.7 l5 "these cases" or "this case"

p.8 l1 "is a currently a hot topic" cut one of the "a"s

p.8 l9 "slope" instead of "slops"

p.9 l5-6 replace "to which" by "in which"

p.9 l14 "growth" instead of "grows" or write "its surface grows quadratically"

---

## Referee Comment (RC2) · Anonymous Referee #2 · 12 Nov 2020

This paper aims to provide observational support in favour of the idea that the wind power input satisfies a fluctuation theorem (FT) in some regions of the ocean. FTs have only appeared recently in the literature and have been useful to justify the physical character of (rare) violations of the second law of thermodynamics. In this paper, it is the wind power input that is treated as the dominantly positive quantity and the analogue of the positive entropy production predicted by the second law, while the negative power input events are seen as the analogue of the rare events seemingly violating the second law. Review of the literature on the subject is pedagogical enough that it can be read and understood with little background on the part of the reader. Overall, the paper is relatively clear and easy to follow, while the analysis appears to

be competently done although short on practical details. The main weakness of the paper, however, is that it appears to devote much time explaining why FTs are useful or important in general, without ever really explaining why they are useful or important in the particular case considered by the paper, namely ocean energetics. The negative power input events are presented as 'extreme' events, but it is unclear to what extent this is justified. Are these events related to the passing by of low- pressure systems that result in occasional reversal of the winds relative to prevailing conditions? The authors emphasise that extreme events are often 'key' for the systems considered (by others), but do not explain why these are key for the system they consider. The paper needs to improve on those aspects as well as on the specific points outlined below before it can be accepted for publication.

General comments

Title: A more concise title would be: Empirical evidence of a fluctuation theorem for the wind mechanical power input in the ocean. I suggest using empirical because the estimation of the power input does not just involve satellite data. The authors need to explicitly state that the mechanical power input is due to the wind, as surface buoyancy fluxes also contributes to powering the ocean.

Aim: Could the authors clarify the precise aims of the paper? Is it intended to contribute to the literature about ocean energetics? If so, the authors should provide some review of the literature about ocean energetics. Is it intended to provide a constraint and metric by which to constrain ocean models? If so, the authors should expand on this some more and explain how one should go about it. Even better would be to repeat the calculations using model outputs where the authors find evidence for a FT to establish whether this would be a useful metric to assess models. As written, it is difficult to understand what issues of interest to the oceanographic community the present results are useful for.

More specific comments

1. Abstract, line 3: 'global satellite observations' may be more specific . Scatterometer wind observations and surface current derived altimeter data.

2. Page 1, lines 15-17: The wind stress also includes a form stress component due to the wind blowing creating negative and positive pressure anomalies on the surface waves

3. Page 1, lines 20-21: The energy exchange is not conservative and most of the mechanical energy is dissipated. I don't understand what that means. Clearly, momentum is conserved and energy is transferred from the atmosphere to the ocean. Part of it goes into avaialbel potential energy to push down isopycnals or suck up isopycnals. Does it go into heat rapidly? Ultimately, sure. What are you trying to say here?

4. Page 2, line 5. 'measure' -> 'estimate' or 'evaluate'. The power input is clearly not measured.

5. Page 2, line 12: 'spacial' -> 'spatial'

6. Page 2, lines 16-17: and conversely, turbulent motion depend also on the mean. Does it matter for the arguments developed here?

7. Page 3, line 7: 'existence of a FT was shown empirically'. 'Shown' sounds like a strong word. Suggested sounds more accurate

8. Page 3, line 13. 'Satellite measurements' not onl. 'discuss their relevance' it is not clear to me that this has really been achieved satisfactorily. This needs to be improved.

9. Page 4, line 21: I find reference to 'shear' somewhat confusing, since power is best understood as the product of a force times displacement by unit time. Why not refer to the wind stress rather than the shear? Moreover, the wind stress is not just due to the shear, it also includes a form stress part due to the wind blow creating pressure positive and negative pressure anomalies on the upwind and downstream sides of sea surface waves.

10. Line 25. May be indicate the value of Cd used for the calculations.

11. Page 4, linear 29. 'goestrophic' - > 'geostrophic'

12. Page 4-5, Lines 31-33. What does it mean physically? Is the power converted into available potential energy or is it dissipated into heat? How does this result justify estimating the wind power input proposed by the authors? Are the overall results sensitive to using the surface velocity or 15 m velocity? The calculations seem easy enough to do that the authors should describe both.

13. Page 6, Lines 19-20: 'This indicates the existence of a large deviation principle' What does that mean? What does that imply? Why is this important or useful?

14. Page 8. Lines 6-8. Why is this useful?

15. Page 8. Lines 7-8. 'Extreme events are often key for the system [. . .]' What does that mean? To what extent are negative wind power input 'extreme' and 'key' for the understanding of ocean energetics.

16. Page 9. Lines 14-26. These last three paragraphs are particularly vague and abstract and not really related to any issues pertaining to ocean energetics. Is it possible to link these to ocean energetics in some way? This paper does not contribute to the theory of FT, so it is unclear why it should speculate on it.

---

## Author Comment (AC1) · 1 Dec 2020

[npg, manuscript]copernicus

picture

amsmath

Answers to both reviewers:

Dear Reviewers,

We are grateful to both reviewers for their corrections and comments as they have

increased the quality of the paper. Please find our detailed answers and corrections to both reviewers comments (reproduced in black) below, written in blue. The corrections performed to the manuscript are given in red and an updated version of the manuscript with the corrections highlighted in red is provided.

The purpose of our work is to extend the theory of FTs which is at the heart of non-equilibrium statistical mechanics for the last 30 years to climate science with interacting components. We give examples of immediate applications but to our understanding, a major benefit resides in the development of a new theoretical framework to further our understanding of the fluctuating interaction between different components of the climate system and their predictability as well as its limits to it. In the previous work we have discussed the application of the concepts (Fluctuation dissipation relation, Fluctuation dissipation theorem and the Fluctuation Theorem) to air sea interaction on the basis of idealized bulk formulas. In the present work we discuss the applicability to data from satellite observations.

Both reviewers are concerned with what concrete benefits FTs provide to the understanding of air-sea interaction and climate science. To address this concern we did several changes in the manuscript (see detailed answer below) and also changed the last two paragraphs of the paper which now read:

Finally, we put the theory of FTs in the more general context of climate dynamics. A measurement, especially when coming from satellites always contains some averaging in space and time. A FT, when it applies, will help to relate averages over varying periods and is a powerful tool to guide the up and down-scaling of observational data in time and obtain the statistical information on shorter and longer time scales, which are not explicitly observed. More precisely, when the pdf of the power supply, and therefore also the symmetry function is known form observations for given averaging times the symmetry function can be calculated for shorter and larger averaging times and constrains "half" of the pdf. This is useful in down-scaling and the construction of statistical parameterizations of not directly observed dynamics over shorter time scales. On the
other hand, the information can be useful for developing models for the persistence of events over large time-scales not yet observed. A FT can help to decide if the persistence in time of a phenomena is within the likeliness of the statistically stationary dynamics or due to external influences. Furthermore, when data from observations follow (or not) a FT, model data should do likewise. As such, the FT becomes a tool of investigating the fidelity of models.

We conclude by looking at our results from the stand point of dynamical systems. Statistical mechanics of systems in equilibrium are described by the Boltzmann distribution, which is completely determined by the temperature. In non-equilibrium statistical mechanics no such universal distribution is known (see i.e. **?, ?** and **?**), but some quantities in some processes seem to follow a FT which constraints the pdf and might indicate some universality. The mechanical power-input to the ocean by air-sea interactions, as a forced and dissipative dynamical system, may thus belong to a class of particular non-equilibrium systems exhibiting a FT symmetry property and offer guidance for climate studies.

Furthermore we like to mention that the applications of FTs in climate science are just beginning and other applications will possibly arise. In the present work we base our investigation on previously published theoretical / numerical investigation which show the existence of a FT in power-supply to the ocean in idealized models. We used a 24 years time series at 6h resolution and have just enough data to start seeing FT like behavior. But to our understanding, there is no doubt that climate science is looking towards a rapid increase (in quantity and quality) of available data and the question about the presence of FT like symmetries in the data can be answered more decisively, also for different variables than the mechanical energy input into the ocean. This allows to analyze environmental data based on theories developed in non-equilibrium statistical mechanics for the last 30 years. We are therefore convinced that the reviewers concerns about the immediate benefit of FTs for ocean science concerning prediction and modeling of quantities will disappear automatically with time.

[Figure]

Sincerely,

Achim Wirth, Bertrand Chapron

Anonymous Referee #1

This study investigates empirically whether or not the time integrated input of mechanical power from the atmosphere to the ocean obeys a fluctuation theorem. If this were the case, observations of the very common case where momentum is transferred from the atmosphere to the ocean could be used to infer probabilities for the rare opposite case. The paper is overall well-written and easy to follow, even if the reader is not closely familiar with ocean dynamics or fluctuation theorems. The core idea is sufficiently interesting for publication in this journal and constitutes a natural next step after the first author's previous study of conceptual models (Wirth 2019). The results appear to be somewhat inconclusive but this fact alone should not exclude the paper from publication. I am mainly concerned with the data analysis in section 5 which is not very clearly presented, both in terms of the methodology and the actual discussion and plots.

Specific comments:

p.4 l4-5 "fixed surface area" this is probably not very important but is the surface area actually fixed when the sea state can change over time? If you always consider fixed geographical regions, wouldn't calm conditions lead to a smaller surface area than rough seas?

The roughness of the surface is not considered here. I know changed to:

(the area which spans $10^o$ in the longitudinal and the latitudinal direction)

p.2 l22 "the focus" please make it clear whose focus you mean (the focus of most current research?)

I now changed to:

Furthermore, the research interest in many natural systems lies mostly in the fluctua-
tions rather than in an average state, [...]

p.2 l33-34 "not only concerned with instantaneous values" if I understand correctly, eq.3
doesn't refer to instantaneous values at all, right? In that case you should cut "only"
here.

Done.

p.3 l30 please make it unambiguous that the limit of large $\tau$ relates to both conditions
and not just (ii). Also this is the first instance where $tau_0$ occurs, please explain what
this refers to.

It is now changed to:

The Galavotti-Cohen fluctuation theorem (called FT in the sequel for brevity) holds for
$\mathcal{P}$, if for averaging times larger than a characteristic time scale of the system ($\tau \gg \tau_0$),
two conditions are satisfied: (i) the symmetry function depends linearly on the variable
$z$, and (ii) on $\tau$:

p.5 l27f consider including a map of the world showing these four regions to give non-
oceanographers at least some idea where they are located, how large they are and
what factors might influence the different dynamics.

I did have a world map in a preliminary version of the paper, but the areas are rather
small and not instantaneously visible. The solution is to put at least two maps, one
for the North Atlantic and one for the North Pacific, but this takes too much space in
my rather short paper and also I do already have many figures. Furthermore it is less
the areas than their dynamic regimes which are important, which asks to include some
current / wind information, which asks for individual zooms of the areas. Putting this
might suggest that a FT can be eye-balled, which is of course not the case. When I
give a talk on the subject I point towards the areas on a map, and show films of the
current and the wind data considered, which resolves the problem. I therefore ask to

keep as is. It is a personal preference and other choices are clearly possible.

p.5 l27f do you have some idea how sensitive your results are to the specific choice of your domains?

We show that the FT "works" in the re-circulation areas considered and that it does not work in the turbulent extensions of western boundary currents. It is written in the paper that: "During data analysis, we also found that a FT does not apply when islands or coastlines are present (not shown here). Departure from a FT for the power input to the ocean is found where horizontal dynamics dominates over the vertical ocean-atmosphere momentum exchanges."

Furthermore the analysis is very demanding in computer time which prohibits general investigation.

p.6 l1 what exactly do you mean by "an interval that spans twice the mean value [. . .] from the origin"? 0 +/- 2*mean( $E_tau$ ) ? In that case why is zero not at the center of the left parts of Fig.1-4 ?

We now replaced "mean" by variance. Not all the data obtained is shown in the graphs as the the averages over shorter time have a much lager variance. We adapted the range in the figs to have a good compromise showing the wide pdfs of short averaging and the narrow pdfs of the long averaging. Note that a convergence of the symmetry function is obtained for the limit in taking the long averaging times. Figures 1-4: Please add axis labels to both parts of the figures. Then the captions of Fig. 2-4 don't need to repeat that of Fig.1, "as Fig.1 but for case XY" would be sufficient. Please give the unit of the averaging time as well.

Axises are now labeled. And we added in the legend of the first figure:

The variable $\tau$ gives the length of the averaging interval in terms of observations done every 6 hours.

p.6 l11you state that you will verify Eq. 3 in two steps so the reader expects these

two to be addressed in order. It is however unclear to me which of the following two paragraphs is supposed to refer to which aspect (see further comments below).

We now added:

That is, we first have to confirm that the lines in the right panels of figs. **??, ??, ??** and **??** converge towards straight lines for increasing averaging periods and second we see if the lines superpose when increasing averaging periods.

p.6 l12 you claim that you "determine the slope" but that that slope is never actually shown or discussed directly. Why not fit lines to your curves and show us the estimated slopes (see comment below)? In that way we could also compare whether or not the slope differs between the regions which is hardly possible by comparing curves indifferent plots with different y-axes.

There are already many lines the figures and adding lines makes the figures difficult to see. Furthermore in exps. GSE and KUE the behavior clearly fails to be linear, so lines can not be included. I choose to define the index gamma to investigate linearity. We do not give the value of the slope as we do not have a theory for the slope and how it is related to the dynamics. This is the case in all references on Fluctuation theorems obtained from experiments with turbulent fluids we know of (see e.g. **?**). We have of course tried to find a relation but did not succeed. Please note that it was and is written in the paper: "The contraction rate $\sigma > 0$ see **?, ?, ?** and **?**) depends on the problem considered."

I our case it is influenced by the relation of the average wind to wind variability on different time-scales, the small scale turbulence in the boundary layer and the temperature stratification in the atmosphere.

We now added:

We did not manage to determine it from observed quantities.

p.6 l13 you again mention $tau_0$, can you at least give some rough estimate how long
that time-scale might be, relative to the length of your time series? Could this be inferred from the power-spectrum of the time-series?

The reviewer is right, an estimate of $tau_0$ should be given, but we do not have enough data to provide such a solid estimate. To consider FTs huge amount of data is necessary, which is often not available yet in environmental sciences. In the present work we base our investigation on previously published theoretical / numerical investigation which show the existence of a FT and we have enough data to start seeing FT like behavior. Please note also that $tau_0$ strongly depends on the tail of the pdf, the rare negative events. Results indicate that in the cases where we observe a FT the symmetry function converges to a strait line in about 1 year. The power-spectrum gives information about the amplitude of a given frequency, but the phase is equally important to determine the occurrence of the high amplitude events (in the same manner as phase is important to determine coherent structures in turbulence). So the connection between the power-spectrum and $tau_0$ is subtle.

I now added:

For the extension of the domains within the recirculation area of the subtropical gyre a convergence towards a linear variation with $z$ is observed in less than $t_0 \approx 1$ year.

p.6 18f "This indicates the existence of a large deviation principle" isn't it more important that this convergence is predicted by the FT? What is the relationship between the existence of an LD principle and a FT? Also is this the first or the second part of the verification mentioned above?

The relation of FT and large deviation principal is often asked when I communicate about this work and I wanted to clarify the point here. If the LD exists for all z than the normalized symmetry function converges, but not necessarily to a straight line. So (ii) indicates the existence of a LD (but does not proof it), even if (i) does not hold. So the way it is said in the text is correct. I do not know how to say it correctly in a different and clearer way. If this sentence about LD leads to confusion it can be taken away.

The rest of the paper is completely independent of it. I would prefer to keep it. We now changed :

For the domains within the recirculation area (ASG and PSG) of the subtropical gyre a convergence towards a linear variation with $z$ is observed in less than $t_0 \approx 1$ year. This points towards the existence of a FT, as both points put forward at the beginning of the previous paragraph are observed. For the extensions of the western boundary currents (GSE and KUE), the convergence does not achieve a linear behaviour of the normalised symmetry function. This shows that a FT does not hold, as the first point put forward at the beginning of the previous paragraph is not satisfied.

p.6 l19f "extension of the domains within ...", "extension of the western boundary current" please refer to the different regions by the acronyms you established before and also refer to the figures in which these results are shown.

Done

p.7 l1f I'm not sure why you chose to quantify the linearity of your curves by this specially designed index. If I understand correctly, the scaled symmetry functions corresponding to long averaging times should be linear across the whole range of z-values. Why not simply fit a line via least squares to calculate the overall slope? Use RËȨ2 to get an idea of the goodness of fit and plot the slopes against tau to observe the convergence behavior. I understand that the statistical interpretation in terms of confidence intervals is questionable but I don't see why your index is more appropriate. Unless I misunderstood your definition, there are many non-linear curves for which gamma=1.

Yes, there are non-linear curves for which gamma=1. One known scenario when the FT fails is due to boundary conditions as briefly mentioned in the text. In this case there is a transition in the slope from high to low values, as we observe in our data. Based on this analytically explained scenario I choose gamma the way I did, other choices are clearly possible.

p.8 l7 "extreme events are often key" of course extreme events in general are interesting but your framework doesn't describe just any kind of weather extreme but specifically unusually small (negative) values of atmosphere-ocean momentum transfer. Can you explain a bit more specifically why a rare event wherein the wind in the atmosphere is sped up by the ocean is of interest?

The reviewer is right. We now added:

Extreme negative events lead to strong transfer of energy to small-scale turbulence in the atmospheric and oceanic boundary layers, potentially causing strong mixing in the atmosphere and ocean.

p.8 l9f I like this example, perhaps it would be even more illustrative if you put in actual numbers for tau? Say one month or one year? This, however raises the question how large tau has to be for the FT to hold...

We now added:

The variable $\tau$ gives the length of the averaging interval in terms of observations done every 6 hours, that is $\tau = 400$ corresponds to a period of 100 days. A FT represents a tool to obtain the rare negative events from frequent positive events for all averaging times $\tau > \tau_0 \approx 1$ year

p.8 l12 "all averaging times" if I understand correctly, your FT only makes statements about long averaging times, right?

We now added:

$t > t_0 \approx 1$ year

p.9 l3 "exp2 & 4" please refer either to the figures or the abbreviations of the different, regions in a consistent manner, the terms "expN" were never explicitly introduced.

Oups, yes, now corrected.

[Figure]

p.9 l18 "guide the up and down-scaling" can you either give a reference for this claim or explain a little more how the FT could help with that?

We now added:

More precisely, when the pdf of the power supply, and therefore also the symmetry function is known form observations for given averaging times the symmetry function can be calculated for shorter and larger averaging times and therefore constrains "half" of the pdf. This is useful in down-scaling and the construction of statistical parameterizations of not directly observed dynamics over shorter time scales. On the other hand the information can be useful for developing models for the persistence of events over large time-scales not yet observed.

Technical corrections:

p.2 l14: case mismatch between "the importance [...] is, [...] their imprint", please re-formulate

Done.

p.2 l17-18: the sentence with "can not be understood or modelled" is repeated verbatim, please cut or re-formulate.

Done.

p.2 l32: replace "i.e." by "e.g."

A negative event is when the ocean loses energy, so I would like to keep "i.e." meaning: "that is".

p.4 l7: replace "is" by "should be"

Done.

p.5 l6f "the production has been performed of ..." confusing sentence, do you mean "a near real-time data set, as well as a 24 year reanalysis, [...], have been produced" ?

Done.

p.5 l15 25 or 24 years ?

Now corrected. The ocean data is 25 years but the overlap with the atmospheric data is only 24 years.

p.5 l20 "6h in time and $1/4^o$ in space" this is repeated from the previous sentence.

Yes, but the first time it considers the atmospheric data and the second time it is the atmospheric and oceanic data. We put it to emphasize that both are available at the same resolution in time and space. We now write:

[. . .] at the same resolution in space and time.

p.5 l24 ", For" either change to lower case or start a new sentence

Done.

p.5 l30 "from" instead of "form"

Done.

p.7 l5 "these cases" or "this case"

Done.

p.8 l1 "is a currently a hot topic" cut one of the "a"s

Done.

p.8 l9 "slope" instead of "slops"

Done.

p.9 l5-6 replace "to which" by "in which"

Done.

p.9 l14 "growth" instead of "grows" or write "its surface grows quadratically"

Done.

Anonymous Referee #2

This paper aims to provide observational support in favour of the idea that the wind-power input satisfies a fluctuation theorem (FT) in some regions of the ocean. Fts have only appeared recently in the literature and have been useful to justify the physical character of (rare) violations of the second law of thermodynamics. In this paper, it is the wind power input that is treated as the dominantly positive quantity and the analogue of the positive entropy production predicted by the second law, while the negative power input events are seen as the analogue of the rare events seemingly violating the second law. Review of the literature on the subject is pedagogical enough that it can be read and understood with little background on the part of the reader. Overall, the paper is relatively clear and easy to follow, while the analysis appears to be competently done although short on practical details. The main weakness of the paper, however, is that it appears to devote much time explaining why FTs are useful or important in general, without ever really explaining why they are useful or important in the particular case considered by the paper, namely ocean energetics. The negative power input events are presented as 'extreme' events, but it is unclear to what extent this is justified. Are these events related to the passing by of low- pressure systems that result in occasional reversal of the winds relative to prevailing conditions? The authors emphasise that extreme events are often 'key' for the systems considered (by others), but do not explain why these are key for the system they consider. The paper needs to improve on those aspects as well as on the specific points outlined below before it can be accepted for publication.

Concerning the lack of concrete applications of FTs in air-sea interaction please see my answer to both reviewers in the beginning of this reply

General comments

Title: A more concise title would be: Empirical evidence of a fluctuation theorem for the wind mechanical power input in the ocean. I suggest using empirical because the estimation of the power input does not just involve satellite data. The authors need to explicitly state that the mechanical power input is due to the wind, as surface buoyancy fluxes also contributes to powering the ocean.

We agree and changed the title to:

Empirical evidence of a fluctuation theorem for the wind mechanical power input into the ocean

Aim: Could the authors clarify the precise aims of the paper? Is it intended to contribute to the literature about ocean energetics? If so, the authors should provide some review of the literature about ocean energetics. Is it intended to provide a constraint and metric by which to constrain ocean models? If so, the authors should expand on this some more and explain how one should go about it. Even better would be to repeat the calculations using model outputs where the authors find evidence for a FT to establish whether this would be a useful metric to assess models. As written, it is difficult to understand what issues of interest to the oceanographic community the present results are useful for.

Please see my answer to both reviewers in the beginning of this reply. Performing the same analysis on model data is planned, but this is another paper. Here we want to discuss the existence of FTs in observations. We added in the introduction:

For a general discussion on air-sea interaction we refer to ?, for ocean energetics to ? and for wind work to ?.

More specific comments

1. Abstract, line 3: 'global satellite observations' may be more specific . Scatterometer wind observations and surface current derived altimeter data.

Yes, but then there is also drifter data and in-situ measurements. We are afraid being at the same time to specific and not specific enough in the abstract. We prefer writing that the basis are 'global satellite observations' and being more specific in the Data section and most importantly referring to the work were this rather involved products are described in all detail. Other choices are clearly possible.

2. Page 1, lines 15-17: The wind stress also includes a form stress component due to the wind blowing creating negative and positive pressure anomalies on the surface waves

By shear we mean the difference of the wind and the currents near the surface. In the present paper we are not concerned with the details of the air-sea interaction at small scales but suppose that these are parameterized by bulk formulas. That is why we write : [...] due to the difference between the atmospheric winds and the ocean currents near the surface in the corresponding planetary boundary layers." and not "at the surface". We now added:

In the present work we do not discuss the various physical processes occurring at the air-sea interface which are important for the momentum transfer.

We now replace "shear" by "shear-stress" in the text.

3. Page 1, lines 20-21: The energy exchange is not conservative and most of the mechanical energy is dissipated. I don't understand what that means. Clearly, momentum is conserved and energy is transferred from the atmosphere to the ocean. Part of it goes into avaialbel potential energy to push down isopycnals or suck up isopycnals. Does it go into heat rapidly? Ultimately, sure. What are you trying to say here?

In air-sea interaction momentum is conserved but not energy (it resembles an inelastic collision of two objects, that stick together after collision). Most of the energy goes into 3D turbulence in the atmospheric and oceanic boundary layers with a direct energy cascade to dissipation into heat a large part goes into wave generation.

4. Page 2, line 5. 'measure' -> 'estimate' or 'evaluate'. The power input is clearly not measured.

Done.

5. Page 2, line 12: 'spacial' -> 'spatial'

Done. (The dictionary says that spacial is ok too)

6. Page 2, lines 16-17: and conversely, turbulent motion depend also on the mean. Does it matter for the arguments developed here?

It does, but here we want to emphasize the closure problem, that is the large scales we are usually interested in, in climate sciences, can not be modeled without some knowledge of the small scales. We now added:

, and vice versa.

7. Page 3, line 7: 'existence of a FT was shown empirically'. 'Shown' sounds like a strong word. Suggested sounds more accurate

Done.

8. Page 3, line 13. 'Satellite measurements' not onl. 'discuss their relevance' it is not clear to me that this has really been achieved satisfactorily. This needs to be improved.

See our answer to both referees in the beginning of this reply

9. Page 4, line 21: I find reference to 'shear' somewhat confusing, since power is best understood as the product of a force times displacement by unit time. Why not refer to the wind stress rather than the shear? Moreover, the wind stress is not just due to the shear, it also includes a form stress part due to the wind blow creating pressure positive and negative pressure anomalies on the upwind and downstream sides of sea surface waves.

Yes, the reviewer is right but by writing "wind stress" we are afraid that the reader thinks

that we are using the approach where the force is calculated based on the wind only and not the difference between wind and current. This is detailed in section 3. We now replace "shear" by "shear-stress" in the text.

10. Line 25. May be indicate the value of Cd used for the calculations.

We now added:

Variations of the drag coefficient are not considered and all the results are independent of a constant $C_d$.

11. Page 4, linear 29. 'goestrophic' - > 'geostrophic'

Done.

12. Page 4-5, Lines 31-33. What does it mean physically? Is the power converted into available potential energy or is it dissipated into heat? How does this result justify estimating the wind power input proposed by the authors? Are the overall results sensitive to using the surface velocity or 15 m velocity? The calculations seem easy enough to do that the authors should describe both.

The wind injected at the surface goes into waves or is dissipated locally in the Ekman layer (see Zhai et al.), has no direct significance on the ocean dynamics. This why we did not consider it here.

13. Page 6, Lines 19-20: 'This indicates the existence of a large deviation principle 'What does that mean? What does that imply? Why is this important or useful?

The relation of FT and large deviation principal is often asked when I communicate about this work and I wanted to clarify the point here. If the LD exists for all z than the normalized symmetry function converges, but not necessarily to a strait line. If this sentence about LD leads to confusion it can be taken away. The rest of the paper is completely independent of it. I would prefer to keep it.

14. Page 8. Lines 6-8. Why is this useful?
If a FT holds we have "half of the pdf" in the case of non-equilibrium stat. mechanics where we do not know the pdf this is the only information we have and it is useful. This is now discussed in more detail in the Conclusions (see answer to both reviewers above).

15. Page 8. Lines 7-8. 'Extreme events are often key for the system [...]' What does that mean? To what extent are negative wind power input 'extreme' and 'key' for the understanding of ocean energetics.

They are extreme because they are in the tails of the pdf. In this events, both, the atmosphere and the ocean loose energy, so large amounts of energy go into small-scale turbulence. We now write:

Extreme negative events lead to strong transfer of energy to small-scale turbulence in the atmospheric and oceanic boundary layers, potentially causing strong mixing in the atmosphere and ocean.

16. Page 9. Lines 14-26. These last three paragraphs are particularly vague and abstract and not really related to any issues pertaining to ocean energetics. Is it possible to link these to ocean energetics in some way? This paper does not contribute to the theory of FT, so it is unclear why it should speculate on it.

We consider if FT is applicable to air-sea interaction and find that is does in some cases. These last three paragraphs are key as they show how FTs can be useful and the last paragraph puts the work in a larger context, it does not speculate. So we would like to keep the paragraphs. We rewrote the last three paragraphs (see answer to both reviewers above).

---

## Editor Comment (EC1) · Balasubramanya Nadiga (Editor) · 3 Dec 2020

I think that the (frequent) usage of "extreme" event(s) by the authors in the manuscript is misleading considering that they are analyzing the probability of rare, practically speaking small negative entropy events.

As well, characteristic time of a system is something that there would generally be consensus on even if there are various measures of it. (A measure of it can be stated/defined and computed. e.g., a decorrelation time scale, an integral time scale, etc.) The authors' use of "characteristic" time is different. It is particular to the analysis being conducted and in a posterior sense may exist or may not exist. Given this, it

might be better to call it something else unless the authors' want to take the extra steps towards closing the loop through further interpretation and relating it to what something for which there is a possibility of consensus.

---

## Referee Report (RR1)

Review of

Wirth et al.: *"Empirical evidence of a fluctuation theorem for the wind mechanical power input into the ocean"*, submitted to NPG

The paper reports on the application of a thermodynamical law, namely a fluctuation theorem, to the exchange of mechanical power at the ocean surface. Necessary measurement data were obtained from 24 years of global satellite observations.

The reasearch described in the paper is highly interesting and relevant to geophysics. Fluctuation theorems (FT), especially the one discussed in the paper, represent newer results from thermodynamics of non-equilibrium systems. Originally derived for and applied to discrete, microscopical systems, their application to macroscopic systems obeying a continuous description is currently a very active field of science. As FTs extend the second law of thermodynamics from an inequality to an exact equation, strong conclusions can be drawn for systems which allow their application. This is also the case for the paper under review.

The reviewer is not an oceanographer and can not evaluate the oceanographic aspects of the paper. However, the exchange of mechanical energy or power at the interface between atmosphere and ocean appears to be a good candidate for an investigation of thermodynamical aspects. As FTs are valid in a very fundamental sense (comparable to, e.g., energy conservation), the detailed processes in both ocean and atmosphere might even not affect the thermodynamics at the interface very much.

After an introduction, the paper first explains the FT under investigation, namely a Galavotti-Cohen fluctuation theorem, frequently also called a "Detailed Fluctuation Theorem" in contrast to an "Integral Fluctuation Theorem". Next, the derivation of the mechanical power input at the sea-air interface is described, followed by the data used in the study. Results are presented in the next section, showing that the FT is fulfilled in two of the four investigated ocean regions. The discussion section finally draws conclusions from the results.

The methodology and the reasoning of the paper are scientifically sound and well performed. The conclusions drawn in the discussion section highlight the potential of FTs in the applied sciences, such as geophysics. Namely the last two paragraphs of the paper discuss the high relevance for assessment of extreme event statistics (and thus, e.g., climate modelling) and general properties of dynamical systems.

The paper in general, and especially the conclusions, have already benefitted substantially from the previous review process. Publication is recommended. My recommendations for improvements are restricted to few technical remarks, which do not affect the scientific contents:

1. p.2 l.25 Put a comma after "fluctuations"; end the sentence after "state" and begin the next one with a capital "Weather".

2. p.3 l.5 "data" is plural, not singular. Therefore it should be "data come" instead of "data comes". Please also check this in the rest of the paper.

3. p.3 l.31 "chonvergence" → "convergence"

4. p.4 l.9 Consider adding "in turbulence" to the end of the last sentence of the paragraph, as this conclusion is probably restricted to turbulence.

5. p.4 l.30 The mentioned "vector product" is, to my understanding, actually a *scalar product*, which is quite a difference. The power $\mathcal{P}$ should be a scalar quantity.

6. p.10 l.18 "form" → "from"

7. p.10 l.22 "a phenomena" → "a phenomenon", the singular form

---

## Author Response (AR2)

Answers to both reviewers:

Dear Reviewers,

We are grateful to both reviewers for their corrections and comments as they have increased the quality of the paper. Please find our detailed answers and corrections to both reviewers comments (reproduced in black) below, written in blue. The corrections performed to the manuscript are given in red and an updated version of the manuscript with the corrections highlighted in red is provided.

Reviewer 1:

The authors have satisfactorily addressed all points raised during peer review. The "Results" section in particular is easier to understand now and the "Discussion" gives readers a better idea of the potential benefits of an FT. I have only a few very minor remarks that could be dealt with before publication.

p.2 l.3 missing space between "the" and "shear"

Done

Section 4 If you do not wish to include a map of the domains, I will not insist on it. I just wanted to mention that you could theoretically also add a video (as mentioned in your reply) as a supplement if you thought that would help readers understand your ideas better. It is probably not necessary though.

I will make a video of this research work

p.6 l.3 "twice the variance" do you really mean variance or standard deviation? Variance would be a little bit weird as it has units of z squared so this kind of range only makes sense for unit-less variables like yours.

Done

p.6 l.6 "exp. 1" better to use "ASG" as this is the only remaining use of "exp. X"

Done

p.6 l.24 "less than $t_0$" this should be "$\tau_0$", right?

Done

p.9 l. 20 remove the "." after "year"

Done

Thank you!

Reviewer 2:

Review of

Empirical evidence of a fluctuation theorem for the wind mechanical power input into the ocean by Wirth and Chapron

**Summary and recommendation**: The revised version of this paper only differs in minor ways from the previous one and has not really improved. The more I read the paper, the more I find it confusing. Although it is easy to follow, I think that this paper does not sufficiently pay attention to details, does not sufficiently justify or explain its methodology, and does not critically discuss its results and their robustness enough. The following provides a list of what needs to be improved to meet the scientific standard required for publication. All the issues listed should be addressable relatively easily. All is required is that the authors try to put themselves in the shoes of a regular oceanographer.

**Major Issues**

– **Definition of the wind stress**: The first main issue that I find problematic is the authors' claim that the wind power input is dominated by the shear stress component of the wind stress without any form of justification to back up this claim.

To my understanding, I answered this concern in my last reply to this reviewer and made changes in the text to make it more clear. In our work the smallest horizontal scale is several tenths of kilometers. Parameterizations of the meachanical power input to the ocean are based on the difference between the atmospheric winds and the ocean currents near the

surface at large horizontal scales. In this case there is shear-stress at the interface. How the power is actually transmitted at the molecular scale or at the scale of the waves (meter to a few tenths of meters) is another question. At the wave-scale the predominant process is definitely the form stress. Including text which discusses the processes at scales which are not considered in the paper will only lead to confusion. We therefore refere the reader to published work dedicated to this subject.

I also find it problematic that the authors never mention or acknowledge the existence of the form stress due to atmospheric pressures fluctuations on surface waves and swell, Grachev et al. (2003) https://doi.org/10.1175/1520-0485(2003)033%3C240 for instance. As far as I am aware, the 'wave' stress is not in general negligible, as it can cause the wind stress direction to be different from that of the wind and drive an atmospheric jet in the regions of light wind, see Hanley and Belcher (2010). Of course, the authors have the right to retain only the shear stress in their calculations if they want to, but they should explicitly acknowledge that this is an approximation that could potentially invalidate their results.

It was and is written in the paper: "In the present work we are not concerned with the details of the exchange in the respective boundary layers (see e.g. **?**) but suppose that it is well represented through bulk formulas of air-sea interaction (**?**)." The first work presents a detailed discussion on the processes at the wave-scale and below, while the second discusses the details of bulk formulas and gives the important references on the subject.

Moreover, I also checked the paper by Fairall et al. (2006) cited by the authors as the basis for the wind stress calculations and found that the authors' approach seems to be different. Indeed, in Fairall et al., the wind stress is calculated as $\tau = \rho_a C_d S (U_{10} - U_o)$, where $S$ is the average atmospheric wind, not the instantaneous wind relative to ocean surface currents.

Fairall et al. write (page 572, lines: 39 –43): "[...] S is the mean wind speed (relative to the ocean surface), which is composed of a mean vector part (U and V components) and a gustiness part (U g ) to account for subgrid-scale variability:" The averaging referes to the average over the grid-scale as opposed to unresolved subgrid-scale quantities. The bulk formulas give the products of subgrid-scale quantities, averaged over the grid-scale, based on quantities averaged over the grid scale. Every observation includes some avearing in time and space, so does ours, performed every 6h and averaged over a square exending $1/2^o$ in both horizontal directions, our grid-scale. And yes, it is relative to ocean surface currents as imposed by the Galilean invariance of Newton's laws.

Moreover, it is not clear from the paper what Uo the authors are using in their calculation of the wind stress. The formula should be used with the total surface velocity including both ageostrophic and geostrophic component, but the way the paper is written suggests that the authors may have used the 15 meters velocity instead. It is essential that the authors clarify this point.

It was and is written in the paper (end of the "Power Input" section): "In the present work, largely building on 15-m drogued drifter velocities (**?**), we use for $\mathbf{u_o}$ the estimation of the current velocity at 15m depth." Please note that in a stationary Ekman layer the surface velocity is at $45^o$ (const. viscosity) to the shear, while the Ekman transport is at $90^o$. So in this case power is provided to the surface flow but not to the Ekman transport. Straightforward calculations show that that the difference is dissipated by viscosity. Taking the surface velocity would mean including the power-input that is dissipated in the Ekman spiral upon injection. This can be avoided (reduced) by taking the velocity below the Ekman spiral. So we use the ageostrophic component minus the Ekman velocity at the surface and the geostrophic component. For basic calculation on the Ekman-dynamics please see: https://hal-bioemco.ccsd.cnrs.fr/INPG/cel-01134110

We now added:

By taking the velocity at 15m rather than at the surface we exclude the power that is promptly dissipated by viscosity in the Ekman-spiral, but include the power that is supplied to the ageostorpic and geostropic dynamics of the core of the mixed layer.

– **Definition of the wind power input**: The second main issue that I find problematic is the authors' definition of the wind power input. As far as I know, the wind power input is by definition the product of the total wind stress by the total surface oceanic velocity, which can be decomposed as the sum of a geostrophic plus ageostrophic part. I think that the

authors are right that a large fraction of the wind power input goes into mixing the mixed layer and ultimately dissipated into heat, but my understanding is that this is related to the ageostrophic work.

As described above the ageostrophic motion can be split into Ekman transport and the part of the sub-mesoscale motion that is not in geostrophic balance. Taking the velocity at 15m excludes the input that is promptly dissipated in the Ekman layer.

Indeed, Roquet et al. (2011) https://doi.org/10.1175/JPO-D-11-024.1 provides physical arguments for why the work against the geostrophic component should be the one driving the large-scale circulation. Based on previous work, I would therefore expect that the right way to compute the wind power input would be by using the surface geostrophic component of the GlobCurrent product. What is the justification of using the 15 meters current? Moreover, the GlobCurrent product description says that both the Ekman and geostrophic components, as well as their sum, is available at 15m. Which one do the authors use? This absolutely needs to be clarified. Moreover, it is essential that the computations are repeated by using the surface geostrophic velocity and the computation of the wind stress actually proposed by Fairall et al. (2006).

When the geostrophic current is used the power injected to the geostrophic circulation is calculated. This, however, excludes the power supplied to the submeso-scale motion that deviates from geostrophic equilibrium, which is today the center of much research. This is stated by Wunsch (1998 JPO 28, 2332), who writes about taking only the geostrophic part: "Equation (1) fails if there are extensive regions where the dynamics differ from simple geostrophy plus an Ekman layer." Using the 15m velocity prevents (part of) this failure. Furthermore the supply to the geostrophic circulation is important when the global or basin-wide circulation is considered. We look at the power-supply at rather fine resolution to a local region that spans only $10^o$ in both horizontal direction. Furthermore, note that following Roquet et al. (2011), not all of the wind work to the geostrophic circulation is injected locally to the geostrophic circulation but transported laterally by Ekman dynamics and the beaviour at basin boundaries becomes important. Therefore: the local injection to the geostrophic circulation is not injected to the local geostrophic circulation. This makes a regional analysis as we perform it questionable (to our understanding) when applied to the geostrophic flow.

– **Negative power input**: The physical meaning of the negative power input events is unclear. If the computations were done for the full wind stress and the ocean surface velocity, they would correspond to instances where momentum is transferred from the ocean to the atmosphere. These events have received some attention in the literature, e.g., Hanley and Belcher (2010) https://doi.org/10.1175/2010JPO4377.1 but these involve the role of swell in regions of light winds. What do these negative events mean here? The authors claim that strong negative events would cause mixing and turbulence (without providing any reference to back up their claim) However, the authors chose the 15 m precisely to avoid this situation, to focus exclusively on a quantity that drives the large-scale circulation, not the turbulence. Doesn't that suggest that their whole approach is inconsistent?

Negative events, the ocean loosing energy, are not necesarrily associated with the atmosphere receiving mechanical (gridscale) energy. Note that when wind and current are in oposing directions, both media are slowed down, loose mechanical (grid scale) energy. This resemble the inelastic collision of particles, we mentioned in the previous reply to this reviewer. When the horizontal circulation in both media at large scales (our grid scale) looses energy the energy has to go to smaler scale (sub-grid scale) turbulence. I am not aware that this has been studie in the literature and can provide no reference. I do not see any inconsistency in our approach.

**Specific comments**

1. Page 1, Line 1: Abstract 'The ocean dynamics is predominantly driven by the shear-stress between [. . . ]' I don't think that this is true. As stated in my previous review, the form stress due to pressure fluctuations on surfaces well and swell often represents a significant component of the wind stress, which can even modify the direction of the wind stress relative to that of the wind. In any case, why is it important for the argument that only the shear stress be retained in the calculation rather than the full wind stress? The authors need to explain why they don't want to include the wave stress in their calculation.

At the scale of the wave, the transfer of momentum is done by the form-stress, but for this we need (on average) a difference between the ocean current and the wind near the surface. This is what I wanted to explain in my answer to the first report and also above in this answer.

2. Page 1, Line 15: '[...] which is described by the fluxes of mechanical power' Is that the right expression? What is the expression for such fluxes?

The sentence is now changed to: "In the present work the exchange of momentum is considered. More precisely, we investigate the flux of mechanical power into the ocean mixed-layer at the ocean surface."

3. Page 1, Line 17: 'In the present work we do not discuss the various physical processes occurring at the air sea interface which are important' It concerns me that the authors don't feel it is needed to discuss the physical processes relevant to their argument. The authors need to acknowledge the different contributions making up the wind stress and explain and justify why they neglect the wave stress

The sentence is now changed to: Various physical processes occurring at the air-sea interface on a large range of scales in space and time are important for the momentum transfer.

The form-stress on the waves is of course the major part of the total stress at the wave-scale, but it is taken into account in the bulk formulas. This is explained in detail in the works that we cite in the paper. We did not atempt a review of the important physical processes at the air-sea interface at our sub-grid scale in the paper, as it would considerably increase its length and refere to published work.

Please note the citation from Fairall (p 584, 2. colomn line 15-22) "In the wind speed range 0–20 m $s^{-1}$ the major remaining surface physics issue is the influence of surface waves on the fluxes. With present techniques, a huge number of observations will be required to obtain definitive results because of the addition of one or two independent variables. High-quality, routine measurements of wave properties is an important technical challenge, so we must hope for a breakthrough in theory or modeling." In our work we applied *Occam's razor* and did not consider the variation of the drag coefficient with the shear. Other choices are clearly possible.

4. Page 1, Line 22: 'The energy exchange is not conservative and most of the mechanical energy is dissipated' I still do not understand what the authors want to say, and what is the point they are trying to make. The answer does not help.

We say that in air-sea interaction mechanical energy is not conserved.

5. Page 2, Line 24: 'Furthermore, the research interest [...]' The examples of whether and climate seems ill chosen, since there is as much interest in the fluctuations as in the average. Climate is by definition an averaged quantity, and estimating the state of the atmosphere at any point in time is crucial for initialising weather forecasts.

The full citation in our paper reads: "Furthermore, the research interest in many natural systems lies mostly in the fluctuations rather than in an average state, weather and climate dynamics are examples where we focus on the fluctuations of the same system on different time scales. For the weather the time scale of interest is from roughly an hour to a week, for the climate the focus is from tenths to thousands of years and beyond." We did never say that the state of the weather and the climate is not important. Weather forcast is concerned with the evolution of the state of the atmosphere. Climate is an average over a certain time interval. If climate were and average over all the data available there would be no climate change ? I typed on google "climate variability" and got $10^9$ results, "climate fluctuations" gives $7 \times 10^8$ results. We did not invent its importance. We now changed the sentence to: Furthermore, the research interest in many natural systems lies also in the fluctuations not only in an average state,

6. Section Power input. I still don't understand what the authors are doing. The expression differs from that of Fairall et al. (1996), for which the term within the square root only involves the atmosphere wind, not the wind relative to ocean surface currents, see major issue above.

Fairall et al. write (page 572, lines: 39 –43): "[...] S is the mean wind speed (relative to the ocean surface), which is composed of a mean vector part (U and V components) and a gustiness part (U g ) to account for subgrid-scale

variability:" . To my understanding relative to the ocean surface means: $u_{10} - u_o^S$, this is also what follows from the Galilean invariance of Newton's laws.

7. Section 4 – Data. I checked the GlobCurrent product description, and it seems to me that this section does not represent an accurate description. In particular, the GlobCurrent website states that different velocities are provided at either: 1) significant wave height, 2) z=0, 3) z=15m. Moreover, both the Ekman and geostrophic velocity are separately available at z = 0 and z = 15m, which is not acknowledged. The authors need to do a better job at explaining what GlobCurrent actually provides, and which exact quantity they are using, whether it is the geostrophic current only or not.

   We give a reference to the GlobCurrent product description and to the major publications in which it is discussed in detail.

8. Section 6 – Discussion. 'We obtain clear evidence that a FT applies to data within the recirculation' I disagree that this has been scientifically established, because the authors did not test the sensitivity of their conclusions to the various assumptions made. In particular, the authors need to test whether their results still hold if they use the surface geostrophic velocity and the correct expression for the wind stress.

   Our expressions of the wind-stress is correct, as we show above. Concerning the geostrophic velocityplease see discussion above. We now deleted "clear".

9. Page 9: 'Extreme negative events lead to strong transfer of energy to small scale turbulence in the atmospheric and oceanic boundary layers, [. . . ]' What is the evidence backing up such a claim? Is this a fact or a speculation? What do they mean by extreme negative events? Doesn't a negative event correspond to transfer of momentum from the oceans to the atmosphere?

   Strongly negative events, the ocean loosing energy, are not associated with the atmosphere receiving energy. Note that when wind and current are in oposing direction, both media are slowed down, loose energy. So a strong loss of energy of the ocean asks for a strong wind against the ocean velocity. When the circulation in both media looses energy the enegy has to go to smaler, subgrid, scale turbulence. I am not aware that this has been studie in the literature and can provide no reference. I do not see any inconsistency in our approach.

   Please note that "Extreme negative events" had already been changed in the manuscript to " Improbable negative events " following the editors request.

---

## Author Response (AR3)

Answer to reviewer 4:

Dear Reviewer,

We are grateful for your corrections and comments as they have increased the quality of the paper. Please find our detailed answers and corrections to your comments (reproduced in black) below, written in blue. The corrections performed to the manuscript are given in red and an updated version of the manuscript with the corrections highlighted in red is provided.

Sincerely,

Achim Wirth, Bertrand Chapron

Reviewer 4:

In full disclosure, I didn't know of the existence of the fluctuation theorem before reading this paper, so please forgive me if some of the comments below may appear naive or out of place.

I am not opposed to the research presented in this manuscript, but I don't understand who it is addressing. If it is addressed to the GFD community, as the motivations and discussion suggest, more efforts should be made to explain why the FT is important and why should one care about it for the power input into the ocean. Yes, we expect some negative values of power input into the ocean over a certain averaging in time and space, and their probability is linked with those of positive values, but so what? Why are these rare negative events particularly relevant for the dynamics of the ocean? In this manuscript, the discussion of the relevance of FT for ocean dynamics is often hand-wavy or redirected to other articles. From an outsider's point of view, it looks like the method is blindly applied to the power input without explaining what oceanic knowledge is gained from it. Turbulence has an imprint on many oceanic quantities, why focusing on power input specifically?

For a statistical quantity all we can know is the pdf and time correlations. In most applications we only know them very roughly because of lack of data. But for some FTs seem to apply, suggesting some underlying symmetry. This is explained in the paper at several places. We understand the criticism of the reviewer, previous reviewers have made comments in this direction and we have added two paragraphs to the discussion section. Making statements beyond this seems to be not justifiable at the present time. Fluctuation theorems have, in the last 20 years revolutionised molecular dynamics and non-equilibrium statistical mechanics and are just about to be applied to fields outside, to fields where fluctuations are non-thermal (turbulence). FTs are recently looked at in wind-tunnel turbulence (https://arxiv.org/abs/2104.03136v1) and other applications (cited in the paper). My focus is on air-sea interaction for which I have recently published a series of theoretical papers (see refs. in the paper) and the present paper is a link of the theory to observations. If we want to understand the turbulent dynamics of the ocean we have to start with the forcing, which is to my understanding (one of) the weak point(s) in ocean modelling. This includes knowledge about the variability of the forcing. But most important: huge amounts of data are available at the ocean surface. This huge amount is barely enough to discuss the existence of FTs.

We now added to the discussion section:

Recently there is an increasing interest in the variability of atmosphere and ocean dynamics and in the exchange of the two, also at high frequency and their contribution toward the lower-frequency air-sea momentum and energy fluxes (e.g., Zhai et al. (2012), Wirth (2021)). It was found that higher frequency wind forcing increases the mixed layer depth (Zhou et al. (2018)). There is evidence that long term variability of the atmosphere ocean system as the Madden-Julian Oscillation and El Niño-Southern Oscillation, are influenced by higher frequency wind forcing (e.g., Bernie et al. (2007), Terray et al. (2012)). FTs when they apply give a connection between events averaged over different lengths of time and can help to evaluate the impact of not explicitly resolved scales. When they do not apply they might help to identify a specific, non stochastic, mechanism responsible.

Some more specific points are detailed below:

On page 6, the authors argue that the pdfs become more centered around unity as tau increases. I don't see this. I would argue that they all converge somewhere between 0 and 1, why is this?

The reviewer is right, looking at the figures this is not clear. When averaged over all data the pdf is a $\delta$-function at 1, by definition. The convergence is at a slow rate, the variance decreases $\sim \tau^{-1/2}$ and data is presented at constant time increments

$\Delta\tau = 125$ for increasing $\tau$. If one looks at graphs of $\tau = 125, 250, 500,$ and 1000 days (constant ratio) it is obvious that "the pdfs become more centered around unity as tau increases". We now added:

With increasing averaging period, the pdfs become more centred around unity, which is the average value, see eq. (**??**). This can be verified by comparing the graphs for $\tau = 125, 250, 500,$ and 1000 days. It is a consequence of the central limit theorem and occurrences of negative values become less likely for larger $\tau$.

Why is the z-axis not always the same for left and right panels, and why isn't the z range not the same for the curves in each panel? For example, in Figure 1, p(z,tau) 1250 has values between -1 and 1 (pink curve in the left panel). Yet, S(z,tau)/tau has values only until 0.5, shouldn't S be defined in the range [0 1]?

In the right panel the values for positive and negative values of the pdfs at $z$ are compared, so only points can be calculated if data is available for both $\pm z$. With increasing averaging time the values for $z < 0$ become less likely and so the analysis can no longer be performed. So if there are only values until 0.5 of S for larger values of $\tau$ this means that for $z < -0.5$ we do not have enough data to calculate S.

Page 9 lines 14-16, the authors argue that improbable events are key for the ocean without explaining why. A very simplified consensus in the literature is that most of the mechanical energy input into the ocean comes from the wind while most of the energy dissipation comes from interaction with topography. Are the authors arguing that a non-negligible part of this dissipation occurs via the interaction with the atmosphere itself?

Yes, surface friction can also deplete energy. This has recently been shown to be important at small scales (in space). Here we only consider temporal behaviour, as to the best of our knowledge no FT exists for fields (spatially extended domains). We now added:

The rare events when the ocean looses energy have recently been the focus of dedicated research (see e.g., Zhai et al. (2012), Wirth (2021)).

Top of page 10, "This suggests that an increased energy cascade, in the extension of boundary currents ..." I don't understand this argument. First, why should we expect an energy cascade in the extension of boundary currents? Is it a direct or inverse cascade? A baroclinically unstable jet will create eddies in the vicinity of the deformation scale, and these can be recycled in the jet itself or advected away. The resulting energy flux in such a small box around the jet is non-trivial and interactions are most likely dominated by non-local transfers. Second, what is the link between the energy cascade and the likelihood of a negative power input event?

The criticism of the reviewer is correct. We have seen a similar behaviour when a friction is added and a direct energy cascade acts as an eddy friction, this is however speculation. We are now more cautious in our argumentation and changed the sentence to:

This suggests that other processes than air-sea interaction dominate in the extension of the boundary currents leading to a departure from a FT symmetry.

Page 10, paragraph about "FTs in the more general context of climate dynamics". This paragraph is important if the manuscript is addressed to the GFD community, yet it is very superficial and speculative. The authors should make an effort of developing these points in a more precise manner and add examples with relevant literature.

FTs are just starting to be used in domains outside of molecular physics. We definitely think that there is a strong potential in interpretation of environmental data, but we do not want to oversell, so we give hints and suggestions of future applications. The suggestions given are true, but we do not know if they are useful in specific applications and we do not want to hypothesise on these points.

We thank the reviewer for the comments and corrections that have helped to increase the quality of the paper.

**References**

Bernie, D., Guilyardi, E., Madec, G., Slingo, J., and Woolnough, S.: Impact of resolving the diurnal cycle in an ocean–atmosphere GCM. Part 1: A diurnally forced OGCM, Climate Dynamics, 29, 575–590, 2007.

Terray, P., Kamala, K., Masson, S., Madec, G., Sahai, A., Luo, J.-J., and Yamagata, T.: The role of the intra-daily SST variability in the Indian monsoon variability and monsoon-ENSO–IOD relationships in a global coupled model, Climate dynamics, 39, 729–754, 2012.

Wirth, A.: Determining the dependence of the power supply to the ocean on the length and time scales of the dynamics between the meso-scale and the synoptic scale, from satellite data, Ocean Dynamics, 71, 439–445, 2021.

Zhai, X., Johnson, H. L., Marshall, D. P., and Wunsch, C.: On the wind power input to the ocean general circulation, Journal of Physical Oceanography, 42, 1357–1365, 2012.

Zhou, S., Zhai, X., and Renfrew, I. A.: The impact of high-frequency weather systems on SST and surface mixed layer in the central Arabian Sea, Journal of Geophysical Research: Oceans, 123, 1091–1104, 2018.